

# Two new species of fossil *Leggadina* (Rodentia: Muridae) from Northwestern Queensland

Ada J. Klinkhamer* and Henk Godthelp

School of Biological, Earth and Environmental Sciences, University of New South Wales, NSW, Australia
* Current affiliation: School of Environmental and Rural Science, University of New England, NSW, Australia

## ABSTRACT

Only three species of fossil murine have been described to date in Australia even though they are often found in fossil deposits and can be highly useful in understanding environmental change over time. Until now the genus *Leggadina*, a group of short-tailed mice that is particularly well adapted to an arid environment, was only known from two extant species: *L. forresti* and *L. lakedownensis*. Here two new fossil species of the genus are described from sites in northwestern Queensland. *Leggadina gregoriensis* sp. nov. comes from the Early Pleistocene Rackham's Roost Site in the Riversleigh World Heritage Area and *Leggadina macrodonta* sp. nov. is from the Plio-Pleistocene Site 5C at Floraville Station. The evolution of the genus *Leggadina* and the lineage's response to palaeoecological factors is considered. Taphonomy of the two fossil deposits is examined and shows marked differences in both faunal composition of the assemblages and preservation. Description of *L. gregoriensis* and *L. macrodonta* extends the known temporal range of the Leggadina lineage by over 2 million years.

## INTRODUCTION

Rodents in Australia include over 70 living species specialised to fill a range of environmental niches from rainforest to arid areas and arboreal to fossorial habitats (*Godthelp, 2001*). All rodents in Australia are part of the subfamily Murinae. This subfamily is thought to have originated in South East Asia, migrating to Australia by rafting and island hopping, and taking advantage of sea level fluctuations (*Archer et al., 1998*; *Godthelp, 2001*).

The murid genus *Leggadina* (*Thomas, 1910*) belongs to the Conilurini tribe of the subfamily Murinae. This tribe is endemic to Australia and is also regularly referred to as the *Mesembriomys* series (*Misonne, 1969*; *Musser & Carleton, 2005*). Older studies using superseded molecular techniques placed *Leggadina* outside the Conilurini group (*Baverstock et al., 1981*; *Watts et al., 1992*). However more recent studies using DNA sequencing include *Leggadina* within the Conilurini (*Steppan et al., 2005*; *Rowe et al., 2008*; *Nilsson et al., 2010*; *Schenk, Rowe & Steppan, 2013*) which is supported by morphological studies (*Tate, 1951*; *Misonne, 1969*). Within this broad Conilurini grouping, the nearest relatives of *Leggadina* are still unable to be determined as studies using either morphological or molecular data produce different results. The genus *Leggadina* contains

Corresponding author
Ada J. Klinkhamer,
aklinkha@myune.edu.au

two living species: *Leggadina forresti* (*Thomas, 1906*) and *Leggadina lakedownensis* (*Watts, 1976*). Species of this genus are characterised by their enlarged first upper molar and reduced third upper molar, an accessory cusp on the anterior of the first upper molar, forward pointing incisors, narrow but large posterior palatal foramina, and straight (or convex) anterior edge to the zygomatic plate (*Watts & Aslin, 1981*).

Tooth morphology is central to the study of rodent systematics because rodents generally have conservative cranial and skeletal morphology, and molecular data is unable to ascertain relationships between fossil groups (*Tate, 1951*). Molars are largely flat with numerous cusps which act as the dominant occlusal surfaces for the grinding of food (*Misonne, 1969*). The position and presence/absence of cusps is key to the morphological identification of species and is particularly important for the identification of fossil species. The most comprehensive study of molar morphology in African and Indo-Australian murids was conducted by *Misonne (1969)*. More recently, molecular techniques have been employed on extant species to determine relationships between groups, with *Rowe et al. (2008)* producing the most comprehensive molecular phylogeny to date on rodents from Australia and New Guinea. Tooth morphology, however, provides the most useful data for fossil rodent systematics because it seems to be the most informative when testing hypotheses based on molecular datasets, modelling, and in determining species relationships (*Wiens, 2004*).

There have only been three species of fossil murines described from Australia: *Pseudomys vandycki Godthelp, 1988*, from the Pliocene-aged Chinchilla locality in southeastern Queensland, *Zyzomys rackhami Godthelp, 1997*, from the Early Pleistocene Rackham's Roost Site in the Riversleigh World Heritage Area in northwestern Queensland, and *Conilurus capricornensis Cramb & Hocknull, 2010*, from late Pleistocene-Holocene cave deposits in eastern Queensland.

The first species to be described in the present study comes from the Riversleigh World Heritage Area in northwestern Queensland, which preserves a rich diversity of fossil vertebrates in limestone rocks from the late Oligocene to the late Pleistocene and Holocene (*Archer et al., 1989*; *Archer et al., 2006*; *Travouillon et al., 2006*). The Rackham's Roost Site at Riversleigh is a breccia deposit in the floor of a fossil cave situated in Cambrian limestone cliffs overlooking the Gregory River. This cave was inhabited by a population of the Ghost Bat *Macroderma gigas* (*Hand, 1996*). Originally identified as a Pliocene-aged site based on biocorrelation (*Archer et al., 1989*), recent radiometric dating of speleothems associated with fossil remains has indicated the site is more likely Early Pleistocene in age (*Woodhead et al., in press*). Fossils found at this site include small mammals believed to be the prey of the Ghost Bat colony, and occasionally larger animals which are believed to have fallen into the cave (*Archer, Hand & Godthelp, 1991*). Rodent fossils found in this deposit represent at least 12 taxa, namely from the genera *Pseudomys*, *Zyzomys* and *Leggadina* (*Godthelp, 2001*). Prior to this study, *Godthelp (1997)* described one species (*Zyzomys rackhami*) from this site.

Site 5C at Floraville Station in northwestern Queensland is quite different from Riversleigh's Rackham's Roost Site. It contains a lower diversity of animals but a much greater range of body sizes. This deposit consists of sandy riverine sediments suggestive of a billabong or waterhole (*Rich et al., 1991*). Rodent remains are thought to have been accumulated through natural mortality and prey of marsupial carnivores (H Godthelp, 2013, unpublished data). The site is Plio-Pleistocene in age (*Rich et al., 1991*), a period that was characterised by great climatic fluctuations and subsequent unpredictability of resources (*Archer et al., 1998*; *Martin, 2006*). Site 5C contains specimens of the murine genera *Rattus*, *Pseudomys* and *Leggadina*, with *Rattus* being by far the most dominant taxon (H Godthelp, 2013, unpublished data). No fossil rodent taxa have previously been described from Floraville.

The description of new species herein almost doubles the number of described fossil Australian murines and will assist in developing a better understanding on the evolution of the murines in Australia, including their initial migration.

## METHODS

Fossil Australian murid specimens were recovered from northwestern Queensland at the Rackham's Roost Site in the Riversleigh World Heritage Area and Site 5C at Floraville Station. Rackham's Roost fossils were recovered by dissolving limestone breccia in 5% acetic acid. The sandy sediment from Site 5C was washed through fine screens to concentrate fossils which were later extracted under a stereomicroscope. A number of fossils recovered at each site were identified as potentially belonging to the genus *Leggadina*. Twenty-eight upper tooth and maxillae specimens from Rackham's Roost and seventeen upper tooth and maxillae specimens from Floraville were analysed and are denoted by the prefix QM F (Queensland Museum Fossil).

Upper molar and upper maxillae specimens from Rackham's Roost Site and Site 5C were observed, as well as lower molar specimens from Site 5C. The upper molar and maxillae specimens from both sites were confirmed as potential new species of the genus *Leggadina*. Observations of the lower molar specimens from Site 5C indicate they are likely attributable to the genus *Leggadina* based on overall similarities to living species of the genus. However, it is not possible to confidently assign them to the same species as the upper molars since none were found in articulation. For this reason, the lower molar specimens from Site 5C have not been described herein.

Univariate and bivariate analyses were conducted using the statistical software program PAST (PAlaeontological STatistics; *Hammer, Harper & Ryan, 2001*) to confirm that the two proposed fossil *Leggadina* species differ from known living and fossil species of the genus. Univariate analyses were conducted to determine the amount of variance within measurements on both fossil and modern taxa using the Coefficient of Variation (CV). The Coefficient of Variation has been widely used to measure the degree of variation within a sample (*Simpson, Roe & Lewontin, 1960*). However, caution must be taken when using this method because there are a number of external variables that can affect CV scores including small sample size, geographic variation and sexual dimorphism (*Plavcan & Cope, 2001*).

Bivariate plots compared upper molar crown length and width data of *Leggadina* specimens (two fossil *Leggadina* specimens, *L. forresti* and *L. lakedownensis*) with closely related species of 'Australian genera' from node W of *Rowe et al.*'s (*2008*) molecular phylogeny which represents conilurine species most closely related to *Leggadina forresti* (*Zyzomys argurus, Pseudomys australis* and *Notomys fuscus*). *Mastacomys fuscus* was removed from the bivariate analysis because its molar morphology diverges so dramatically in both size and cusp arrangement that the fossil specimens collected from the two Queensland sites clearly do not belong to this genus. The greatest length and width of upper molars were used to determine species identification because molar cusp position is too variable, especially with occlusal wear (*Misonne, 1969*).

Measurements were made at the University of New South Wales on a Wild 5MA stereomicroscope with Wild MMS235 Digital Length Measuring Set (accurate to 0.01 mm) and at the Australian Museum on a Leica MZ95 stereomicroscope with graticule (accurate to 0.05 mm). Measurements were cross-checked to ensure comparability by measuring a subset of specimens on both microscopes. No $M^3$ or a molar row has been discovered for the Floraville *Leggadina*, so bivariate plots for $M^1$ and $M^2$ were used to assess separation of these murine species. *Leggadina lakedownensis* could not be included in the $M^2$ analysis as access to specimens was not possible.

Dental nomenclature used herein follows *Musser & Newcomb (1983)* which uses a simplified serial nomenclature that reduces potential issues of conflicting homologies in the upper molars for muroid rodents (Fig. 1). A Wild M3B stereomicroscope was used during the description of new species. Specimens were photographed using a scanning electron microscope (Quanta 200) housed at the University of New South Wales Analytical Centre.

Abbreviations used in this study are defined as follows: $M^1$ = first upper molar, $M^2$ = second upper molar, $M^3$ = third upper molar. All measurements are in millimeters (mm).

The electronic version of this article in Portable Document Format (PDF) will represent a published work according to the International Commission on Zoological Nomenclature (ICZN), and hence the new names contained in the electronic version are effectively published under that Code from the electronic edition alone. This published work and the nomenclatural acts it contains have been registered in ZooBank, the online registration system for the ICZN. The ZooBank LSIDs (Life Science Identifiers) can be resolved and the associated information viewed through any standard web browser by appending the LSID to the prefix "http://zoobank.org/". The LSID for this publication is: urn:lsid:zoobank.org:pub:41DF9EE4-BF1B-492E-AC00-59992E0C28B4. The online version of this work is archived and available from the following digital repositories: PeerJ, PubMed Central and CLOCKSS.

## RESULTS

### Univariate analyses

Coefficients of Variation for all measurements of the fossil taxa suggest that only one species is present in each fossil sample, with values ranging from 3.23 to 7.80 in the

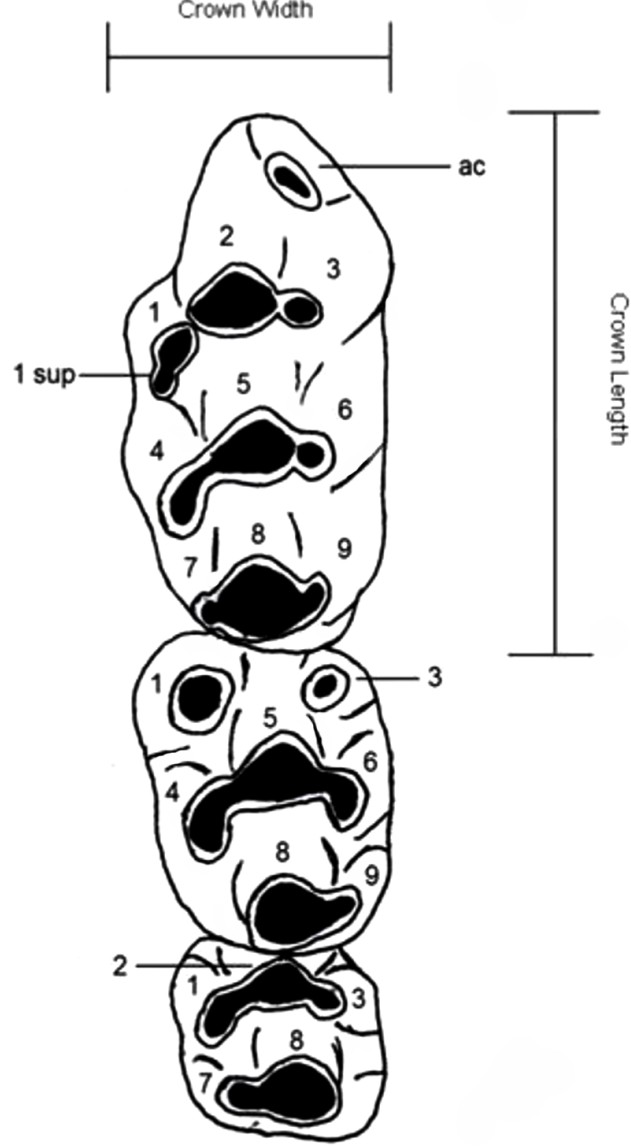

**Figure 1 Dental nomenclature used in the description of fossil *Leggadina*.** Adapted from *Musser & Newcomb (1983)* but modified to better represent features of fossil *Leggadina* specimens. Left upper molar row, cusps (1–9) referred to in text with the prefix 'T', ac, accessory cusp; sup, supplementary. Measurements were taken on maximum crown length and width.

*Leggadina* specimens from Rackham's Roost and 5.50 to 6.06 for the two measurements available for *Leggadina* specimens from Site 5C (Supplemental Information).

## Bivariate analyses

In the bivariate plots, both length and width of $M^1$ and $M^2$ were effective in separating species (Figs. 2 and 3). The $M^1$ plot shows the Rackham's Roost *Leggadina* overlapping with both modern *Leggadina* species (*L. forresti* and *L. lakedownensis*), whereas in the $M^2$ plot, the Rackham's Roost *Leggadina* groups predominately with the Floraville specimens.

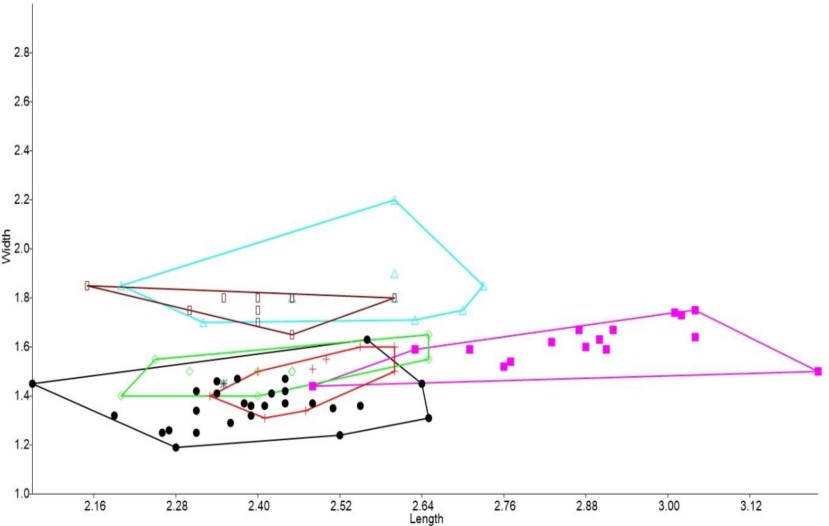

**Figure 2 Bivariate plot comparing M$^1$ between murine species.** Bivariate plot of maximum crown length and width of M$^1$ (mm). *Leggadina forresti*, green diamond; *Leggadina lakedownensis*, blue star; *Leggadina gregoriensis*, black circle; *Leggadina macrodonta*, pink square; *Zyzomys argurus*, red cross; *Pseudomys australis*, blue triangle; *Notomys fuscus*, brown rectangle.

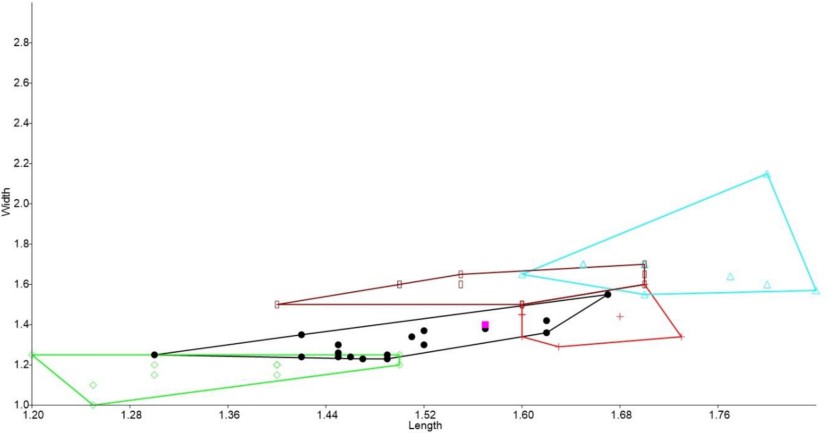

**Figure 3 Bivariate plot comparing M$^2$ between murine species.** Bivariate plot of maximum crown length and width of M$^2$ (mm). *Leggadina forresti*, green diamond; *Leggadina gregoriensis*, black circle; *Leggadina macrodonta*, pink square; *Zyzomys argurus*, red cross; *Pseudomys australis*, blue triangle; *Notomys fuscus*, brown rectangle (*L. lakedownensis* not included).

The Floraville *Leggadina* species distinctly separates from other species based on its greater M$^1$ length. *Pseudomys* and *Notomys* group together in both plots, but separate more in the M$^2$ plot based on length data. In both plots there is a close association between the fossil specimens and *Zyzomys*. More detailed morphological evidence effectively separates *Zyzomys* and the fossil specimens, as detailed in the differential diagnosis.

## SYSTEMATICS

### Differential diagnosis

The fossil species described below refer to the genus *Leggadina* and display characteristics typical of species of this genus. An accessory cusp on the first upper molar is present on all fossil specimens, all upper molars are inclined posteriorly, molar size is reduced along the row, with $M^3$ often half the size or smaller than $M^1$, and the anterior edge of the zygomatic plate is relatively straight (*Watts & Aslin, 1981*). *Leggadina gregoriensis* differs from other species of the genus in the following combination of features: a greatly anteroposteriorly elongated T6 on $M^1$; T1-2 and T4-5 complexes are oriented buccolingually with T3 and T6 swept back at right-angles to lean proximally; an accessory cusp is present but small; $M^1$ is narrow, with $M^2$ and $M^3$ being wider than $M^1$. *Leggadina macrodonta* differs from other species of the genus in the following combination of characters: $M^1$ is enlarged, being approximately 18% larger than *Leggadina forresti* and *L. lakedownensis*; $M^2$ is similarly enlarged, approximately 16% larger than in those species; an anterior cingulum is present and is enlarged with two accessory cuspules that wear to a greatly elongated accessory cusp; T1 and T4 are well-developed and posterolingually aligned; a T1 sup is present on some specimens; the central series of cusps is also enlarged. Both fossil species also differentiate themselves from the two modern *Leggadina* species through the presence of furrows between the lingual and central series of cusps in $M^1$ and $M^2$. Bivariate analyses determined that *L. gregoriensis* and *L. macrodonta* could have been referred to the genus *Zyzomys*. Shared morphological features and differences between *Leggadina* and *Zyzomys* are mentioned here (Fig. 4). *Zyzomys* species often display an accessory cusp on the first upper molar, have a relatively straight anterior edge to the zygomatic plate, and are of similar size to *Leggadina* (*Watts & Aslin, 1981*). A feature clearly distinguishing species of the two genera is a buccal row of cusps present in *Leggadina* species that is absent in *Zyzomys*. A distinctive aspect of *Leggadina* molar morphology, not shared by *Zyzomys*, is the posterior extension of the lingual series of cusps (*Tate, 1951*). For these reasons, the fossil species are referred to the genus *Leggadina* rather than *Zyzomys*. Character states are unable to be discussed here as relationships between the genus *Leggadina* and other murines are uncertain and can change depending on methods used. This situation is not helped by the lack of fossil evidence on murines in Australia.

Superfamily MUROIDEA Miller and Gidley, 1918
Family MURIDAE Gray, 1821
Subfamily MURINAE Gray, 1821
Genus *LEGGADINA* *Thomas, 1910*

**Type species**
*Leggadina forresti* (*Thomas, 1906*)
**Other species**
*Leggadina lakedownensis* *Watts, 1976*

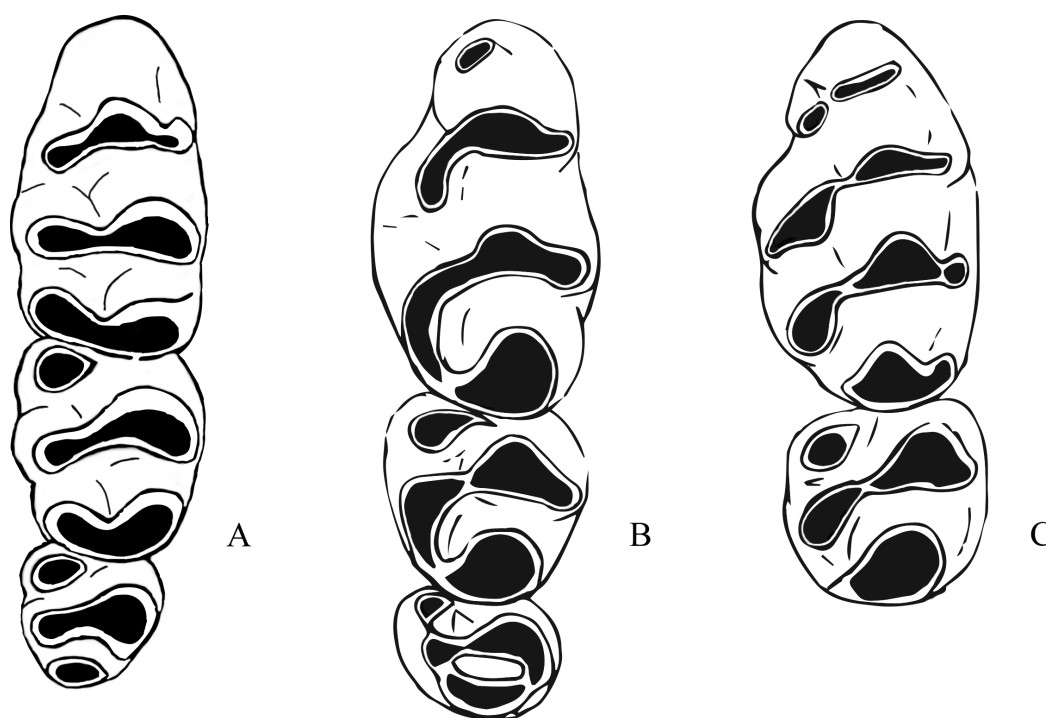

**Figure 4 Morphological differences between fossil *Leggadina* species and *Zyzomys*.** (A) Left upper molar row of *Zyzomys argurus* (*Misonne, 1969*); (B) right upper molar row of holotype (QM F57259) of *Leggadina gregoriensis*, image has been reversed to represent left upper molar row for comparative purposes; (C) left M$^1$ and M$^2$ of *Leggadina macrodonta*, composite of holotype (QM F57276) and paratype (QM F57273). Not to scale.

*Leggadina gregoriensis* sp. nov.

**Holotype**

QM F57259, partial right maxilla with M$^{1-3}$ (Fig. 5).

**Type locality and age**

Rackham's Roost Site, Riversleigh World Heritage Area, northwestern Queensland; Pleistocene (*Woodhead et al., in press*).

**Paratypes**

QM F57244, partial right maxilla with M$^1$ and alveoli of M$^2$ and M$^3$ (Fig. 6); QM F57258, partial left maxilla including zygomatic plate with M$^1$ and M$^2$ (Fig. 7).

**Etymology**

Named for the Gregory River which flows next to the Rackham's Roost Site.

**Diagnosis**

*Leggadina gregoriensis* is characterised by a small accessory cusp, greatly anteroposteriorly elongated T6 on M$^1$; T3 and T6 swept back at right-angles to lean proximally; M$^1$ narrow, M$^2$ and M$^3$ wider.

**Referred specimens**

QM F57240, right M$^1$; QM F57241, left M$^1$ in partial maxilla; QM F57242, right M$^1$; QM F57243, right M$^1$; QM F57245, left M$^1$; QM F57246, right upper molar row in partial

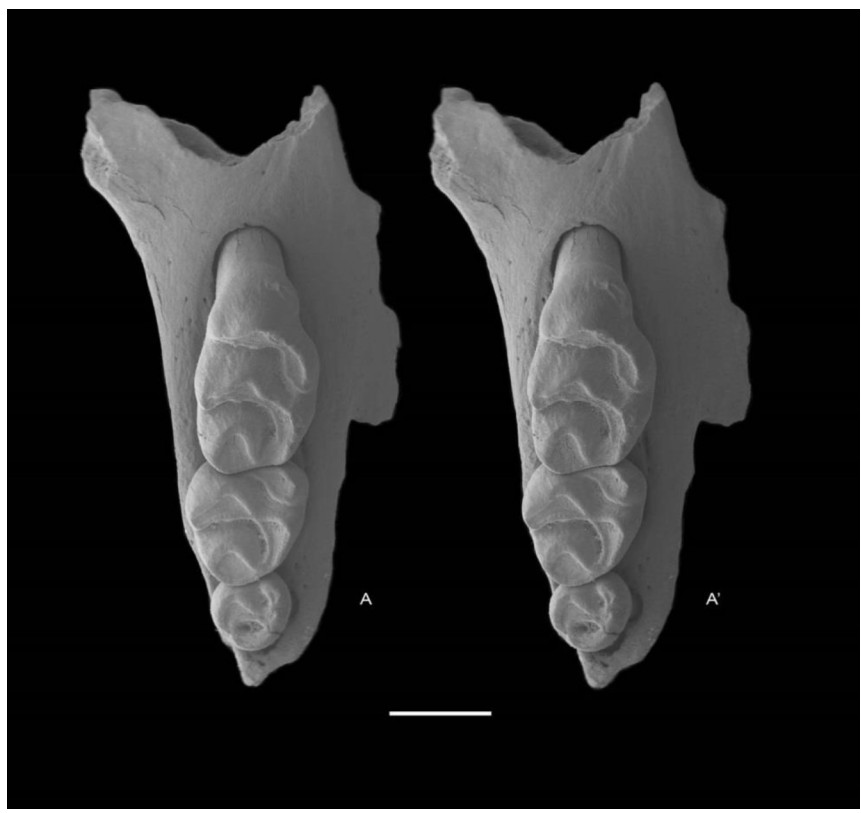

**Figure 5** *Leggadina gregoriensis* **sp. nov. Holotype. QM F57259.** Partial right maxillary with $M^{1-3}$. Occlusal view. $A - A' =$ stereopair. Scale $= 1$ mm.

maxilla; QM F57247, left $M^2$ in partial maxilla; QM F57248, right $M^1$ and $M^2$ in partial maxilla; QM F57249, right $M^1$ and $M^2$ in partial maxilla; QM F57250, right $M^1$ and $M^2$; QM F57251, right $M^1$ and $M^2$; QM F57252, left $M^1$ and $M^2$; QM F57253, right $M^{1-3}$ in partial maxilla; QM F57254, left $M^1$; QM F57255, left $M^1$ in partial maxilla; QM F57256, right $M^1$ and $M^2$ in partial maxilla; QM F57257, left $M^1$ and $M^2$ in partial maxilla; QM F57260, right $M^{1-3}$; QM F57261, right $M^1$; QM F57262, right $M^{1-3}$; QM F57263, right $M^1$ and $M^2$; QM F57264, left $M^1$ in partial maxilla; QM F57265, right $M^1$ in partial maxilla; QM F57283, left upper molar row; QM F39958, left $M^{1-3}$ (Table 1).

## Description

$M^1$ large and elongated. $M^2$ approximately two-thirds the size of $M^1$. $M^3$ smaller again, approximately half the size of $M^2$ (Table 1). Tooth row exhibits spiral torsion, $M^1$ straight with $M^2$ and $M^3$ twisted slightly to the buccal edge. Furrow present between lingual series of cusps and central series of cusps in $M^1$ and $M^2$. Buccal series of cusps reduced along tooth row, central series of cusps enlarged. All cusps inclined posteriorly with tooth crown overlap.

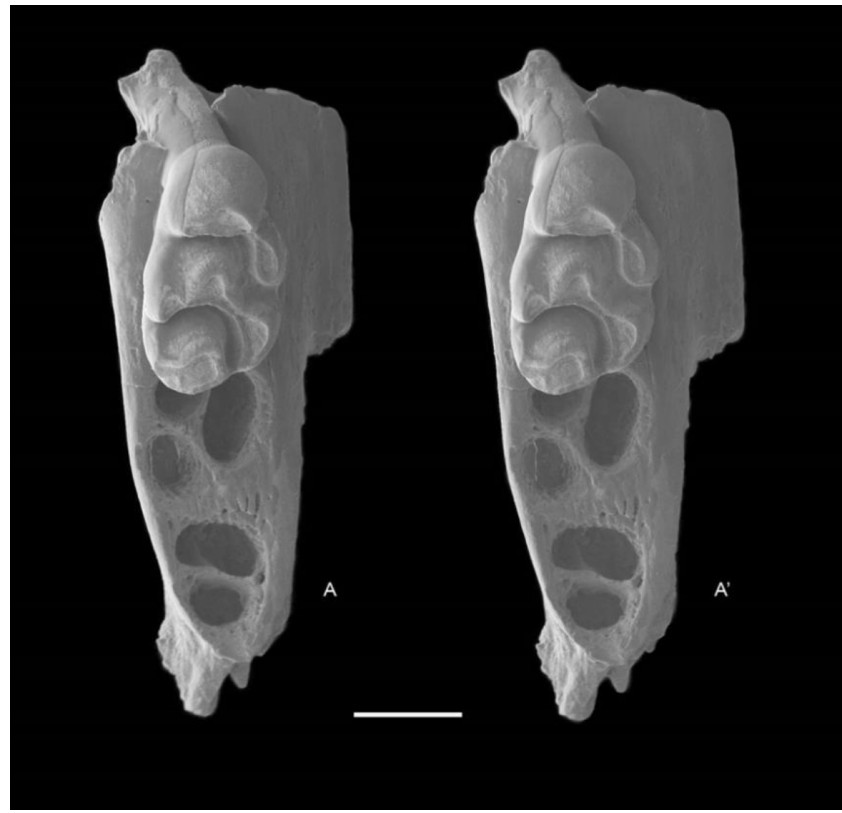

**Figure 6** *Leggadina gregoriensis* **sp. nov. Paratype. QM F57244.** Partial right maxillary with M$^1$. Occlusal view. $A$–$A'$ = stereopair. Scale = 1 mm.

**M$^1$**: Elongated and narrow. Anterior cingulum with a single and small elliptical accessory cusp sweeping backwards along lingual edge. Accessory cusp small in all specimens, almost indistinguishable in QM F57244. T1 very small and circular, connected to T2 at early stages of wear. T2 posteriorly inclined, large and elliptical. It is the highest cusp at early stages of wear but becomes uniform with the other M$^1$ cusps after wear. T1-2 complex buccolingually aligned. T3 positioned to posterior of T1-2 complex, at mid-point of tooth. T3 elliptical, directed proximally and connected to T2 by an enamel rim in the holotype. At early stages of wear it is entirely distinct but merges completely with T1-2 complex after extreme wear. T4 small, circular and merged with T5 at most stages of wear. It sweeps posteriorly from T5 so anterior edge of T4 is in line with the posterior edge of T5. T5 large, subtriangular in occlusal outline and leans posteriorly. Enamel rim connects T5 to both T4 and T6. T6 positioned posterior to T5, elongated anteroposteriorly and directed proximally, similar to T3. T6 merges with T4-5 complex after extreme wear. T6 also distinct from T9 at early stages of wear but merges quickly. Posterior edge of cusps T4–T6 arcs anteriorly to enclose T8. T7 barely discernible in holotype but is present in other specimens at early stages of wear before merging completely with T8. In these specimens it is small and directed posteriorly. T8 very large and circular, directed posteriorly. T9 incorporated at all stages of

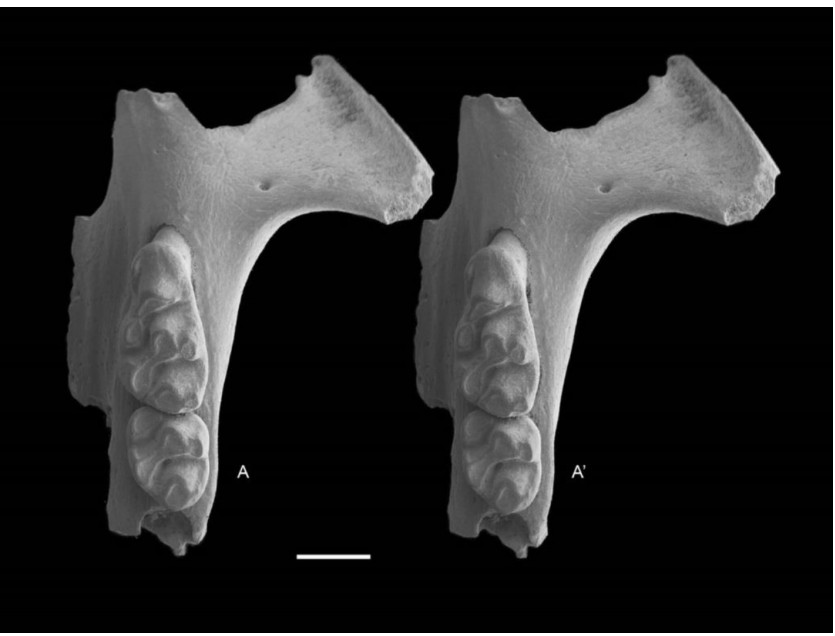

**Figure 7** *Leggadina gregoriensis* sp. nov. Paratype. **QM F57258.** Partial left maxillary including zygomatic plate with $M^{1-2}$. Occlusal view. *A–A′* = stereopair. Scale = 1 mm.

wear with T8. Enamel rim around cusps uniform throughout tooth but becomes slightly wider with extreme wear. Has three roots, all of which directed somewhat anteriorly.

Anterior root largest of the three, circular in shape and positioned under accessory cusp and T1–T3. Lingual root anteroposteriorly stretched, narrow and positioned under T6 and T9. Posterior root smallest of the three, circular and positioned under T8.

$M^2$: Tooth is mostly circular in holotype but shape variable, with other specimens more elongate. Elongation is affected by size of T3 and T8, with the anterior of $M^2$ developing a bulge with increase in T3, similarly, posterior developing a bulge with increase in T8. T1 and T2 absent. T3 distinct and elliptical, directed proximally. T3 and T5 are the highest cusps at early stages of wear but T3 wears faster than T5 to become uniform with the other cusps. T4 small, circular and leans posteriorly. It is incorporated into T5, but also sweeps posteriorly from T5, with anterior edge of T4 in line with posterior edge of T5. T5 subtriangular and directed posteriorly. T6 positioned posterior to T5, elongated anteroposteriorly and oriented proximally. At later stages of wear T6 merges with T4-5 complex. Posterior edge of T4-5 complex and posterior edge connecting T6 with T9 forms anterior arc to enclose T8, similar to $M^1$. T7 absent. T8 large, circular in occlusal outline and directed posteriorly. At extreme stages of wear T8 merges with elongated T6. T9 merges with T8 at all stages of wear, similar to $M^1$. Enamel rim surrounding the cusps of uniform width, becoming thicker with wear.

With three roots, all directed vertically. The anterobuccal and posterobuccal roots of equal size and circular. Anterobuccal root extends from underneath T4 and T5, while

Table 1 Measurements (mm) of *Leggadina gregoriensis* sp. nov

| Specimen no. | M$^1$ | | M$^2$ | | M$^3$ | | M$^{1-3}$ | | M$^{1-2}$ | |
|---|---|---|---|---|---|---|---|---|---|---|
| | L | W | L | W | L | W | L | W | L | W |
| QM F57240 | 2.44 | 1.42 | – | – | – | – | – | – | – | – |
| QM F57241 | 2.65 | 1.31 | – | – | – | – | – | – | – | – |
| QM F57242 | 2.07 | 1.45 | – | – | – | – | – | – | – | – |
| QM F57243 | 2.31 | 1.42 | – | – | – | – | – | – | – | – |
| QM F57244 | 2.37 | 1.47 | – | – | – | – | – | – | – | – |
| QM F57245 | 2.34 | 1.46 | – | – | – | – | – | – | – | – |
| QM F57246 | 2.55 | 1.36 | 1.52 | 1.30 | 1.07 | 1.02 | 4.84 | 1.47 | 3.90 | 1.47 |
| QM F57247 | – | – | 1.62 | 1.42 | – | – | – | – | – | – |
| QM F57248 | 2.44 | 1.37 | 1.62 | 1.36 | – | – | – | – | 3.82 | 1.39 |
| QM F57249 | 2.48 | 1.37 | 1.51 | 1.34 | – | – | – | – | 3.90 | 1.42 |
| QM F57250 | 2.34 | 1.41 | 1.45 | 1.30 | – | – | – | – | 3.79 | 1.45 |
| QM F57251 | 2.56 | 1.63 | 1.67 | 1.55 | – | – | – | – | 4.02 | 1.64 |
| QM F57252 | 2.28 | 1.19 | 1.49 | 1.23 | – | – | – | – | 3.70 | 1.24 |
| QM F57253 | 2.19 | 1.32 | 1.42 | 1.35 | 1.05 | 0.92 | 4.54 | 1.46 | 3.60 | 1.46 |
| QM F57254 | 2.38 | 1.37 | – | – | – | – | – | – | – | – |
| QM F57255 | 2.52 | 1.24 | – | – | – | – | – | – | – | – |
| QM F57256 | 2.39 | 1.32 | 1.47 | 1.23 | – | – | – | – | 3.78 | 1.32 |
| QM F57257 | 2.31 | 1.25 | 1.46 | 1.24 | – | – | – | – | 3.59 | 1.34 |
| QM F57258 | 2.41 | 1.36 | 1.49 | 1.25 | – | – | – | – | 3.81 | 1.36 |
| QM F57259 | 2.26 | 1.25 | 1.30 | 1.25 | 0.85 | 0.85 | 4.29 | 1.37 | 3.51 | 1.37 |
| QM F57260 | 2.27 | 1.26 | 1.45 | 1.24 | 1.00 | 0.95 | 4.45 | 1.39 | 3.65 | 1.39 |
| QM F57261 | 2.44 | 1.47 | – | – | – | – | – | – | – | – |
| QM F57262 | 2.36 | 1.29 | 1.42 | 1.24 | 0.96 | 0.90 | 4.56 | 1.35 | 3.10 | 1.35 |
| QM F57263 | 2.39 | 1.36 | 1.45 | 1.26 | – | – | – | – | 3.70 | 1.39 |
| QM F57264 | 2.31 | 1.34 | – | – | – | – | – | – | – | – |
| QM F57265 | 2.64 | 1.45 | – | – | – | – | – | – | – | – |
| QM F57283 | 2.51 | 1.35 | 1.52 | 1.37 | 1.04 | 1.00 | 4.68 | 1.35 | 3.79 | 1.35 |
| QM F39958 | 2.42 | 1.41 | 1.57 | 1.38 | 0.94 | 0.90 | 4.57 | 1.42 | 3.84 | 1.42 |

**Notes.**
   L, maximum length; W, maximum width.

posterobuccal root positioned beneath T8. Lingual root large and elongated, extending from T3 to T6.

**M$^3$**: Tooth circular with a bulge on anterolingual edge for T3, cusp height uniform. T1 and T2 absent. T3 small, circular and distinct, directed proximally. Furrow between T3 and T4-6 complex ensures T3 distinct in all but very late stages of wear. T4 completely incorporated into T5. It sweeps posteriorly markedly from T5, directed posterobuccally. T5 subtriangular in occlusal outline, large and directed posteriorly. T6 small and subtriangular. It merges with T5, slightly sweeping posteriorly from T5 with enamel rim connecting to T8-9 complex. Posterior edge of T4-5 complex curves anterobuccally, with posterior edge of T6 curving anterolingually. T7 absent. T8 large, elliptical and orientated
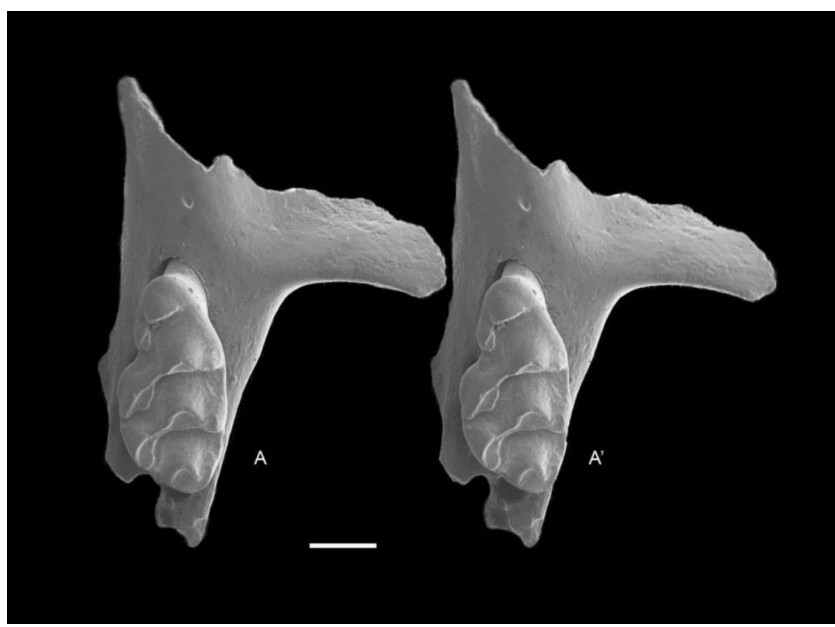

**Figure 8** *Leggadina macrodonta* **sp. nov. Holotype. QM F57276.** Partial left maxillary including zygomatic plate with $M^1$. Occlusal view. $A$–$A'$ = stereopair. Scale = 1 mm.

vertically. Anterior edge of T8 curves posteriorly. Anterior edge of T8 combined with posterior edge of T4-6 complex creates elliptical furrow. T9 entirely incorporated into T8. Enamel rim uniform in width and connecting all cusps except T3 in holotype which only connects at very late stages of wear.

With three roots all directed vertically. Anterobuccal root small and circular, extending from beneath T5. Anterolingual root slightly larger and more elongated than anterobuccal root and positioned under T3 and T6. Posterior root largest of the three, supporting approximately half tooth length and extending from T8.

Attachment node for the origin of the superficial masseter is of moderate size and well defined in some specimens, positioned anterior to $M^1$. Posterior extent of anterior palatal foramen lies at anterior root of $M^1$. Zygomatic plate of QM F57258 wide with posterior edge convex (Fig. 7).

*Leggadina macrodonta* sp. nov.

**Holotype**

QM F57276, partial left maxillary including zygomatic plate with $M^1$ (Fig. 8).

**Type locality and age**

Site 5C, Floraville Station, northwestern Queensland; Plio-Pleistocene (*Rich et al., 1991*).

**Paratypes**

QM F57273, partial left maxillary with $M^2$ (Fig. 9); QM F57268, left $M^1$ (Fig. 10); QM F57275, partial left maxillary with $M^1$ and alveoli of $M^2$ (Fig. 11).

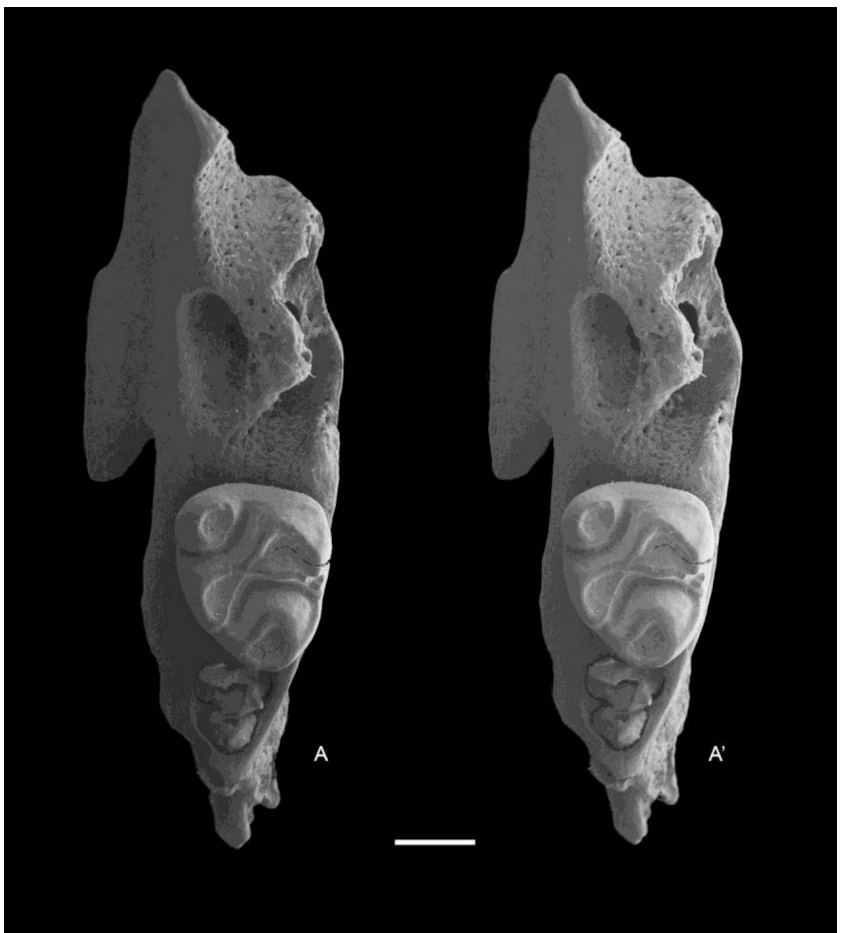

**Figure 9** *Leggadina macrodonta* **sp. nov. Paratype. QM F57273.** Partial left maxillary with $M^2$. Occlusal view. $A–A'$ = stereopair. Scale = 1 mm.

**Etymology**

Named for the distinctively large size of the first upper molar.

**Diagnosis**

*Leggadina macrodonta* is characterised by a greatly enlarged $M^1$ and $M^2$; enlarged anterior cingulum with two accessory cuspules that wear to a greatly elongated accessory cusp; well-developed T1 and T4 posterolingually aligned; enlarged central series of cusps.

**Referred specimens**

QM F57266, right $M^1$; QM F57267, left $M^1$; QM F57269, left $M^1$; QM F57270, right $M^1$; QM F57271, left $M^1$; QM F57272, right $M^1$; QM F57274, left $M^1$; QM F57277, left $M^1$; QM F57278, right $M^1$; QM F57279, right $M^1$; QM F57280, right $M^1$; QM F57281, right $M^1$; QM F57282, left $M^1$ (Table 2).

## Description

Complete tooth row not known. $M^1$ and $M^2$ are isolated specimens, no specimen of $M^3$ found to date. $M^1$ large, $M^2$ approximately half length of $M^1$ (Table 2). Furrow between

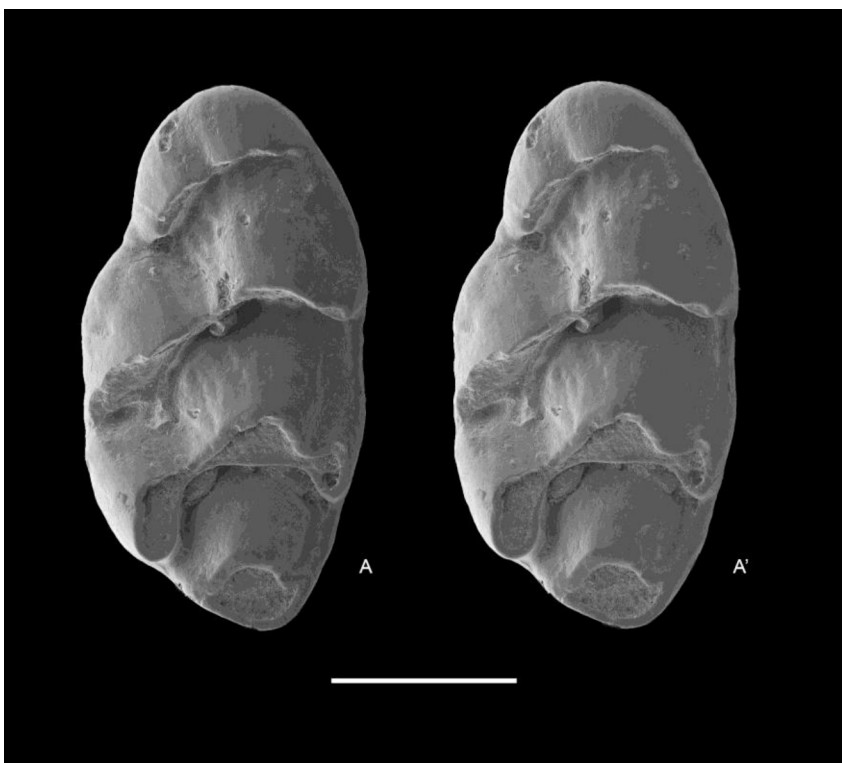

**Figure 10** *Leggadina macrodonta* **sp. nov. Paratype. QM F57268.** Left M$^1$. Occlusal view. $A$–$A'$ = stereopair. Scale = 1 mm.

lingual series and central series of cusps in M$^1$ and M$^2$. Buccal series of cusps reduced in M$^1$, all cusps inclined posteriorly.

**M$^1$**: Tooth elliptical with thin and uniform enamel rim around all cusps. Two small accessory cusplets present on anterior cingulum in holotype. With wear they become one very large accessory cusp, elongated posterolingually, sweeping back along lingual edge. T1 large and elongated, becoming more elongated with wear. Anterior edge of T1 sits posterior to T2, at half-way point of tooth. T1 orientated posteriorly with axis of cusp stretching posterolingually, parallel to single accessory cusp in specimens other than holotype. It merges with T2 at late stages of wear. T1 sup present on some specimens, situated on posterolingual edge of T1. It is small and circular, merging into T1 with wear. T2 of moderate size and subtriangular in occlusal outline. T3 very small and circular, sweeping slightly posteriorly from T2 in some specimens. T3 often connected to T2 by enamel rim, later merging with wear. T4 large and tear-shaped, increasing in size posteriorly with wear but never merging with T7 or T8. It only barely merges with T5, even at late stages of wear. Large size of T4 together with similarly sized T1 creates a bulge on lingual edge of tooth, enlarging width of otherwise slender tooth. Anterior edge of T4 sits posterior to the posterior edge of T5. T4 higher at posterior edge than anterior edge. Cusp posteriorly inclined, with axis running almost parallel to main axis of tooth. T5 large and subtriangular, orientated posteriorly. T6 circular, elongating anteroposteriorly with wear

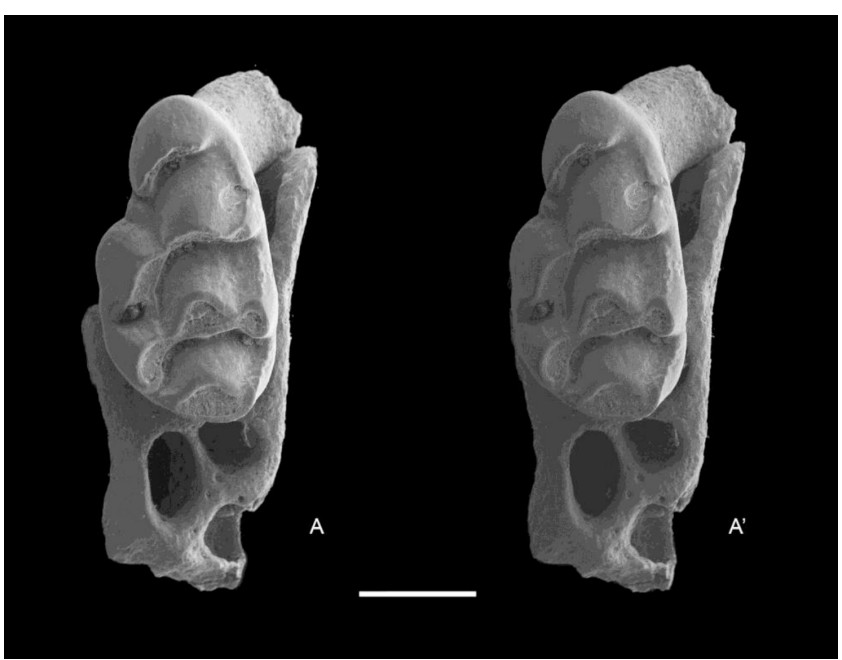

**Figure 11** *Leggadina macrodonta* sp. nov. Paratype. **QM F57275.** Partial left maxillary with M$^1$ and alveoli of M$^2$. Occlusal view. $A$–$A'$ = stereopair. Scale = 1 mm.

**Table 2  Measurements (mm) of *Leggadina macrodonta* sp. nov.**

| Specimen no. | M$^1$ | | M$^2$ | |
|---|---|---|---|---|
| | L | W | L | W |
| QM F57266 | 2.48 | 1.44 | – | – |
| QM F57267 | 3.04 | 1.75 | – | – |
| QM F57268 | 3.04 | 1.64 | – | – |
| QM F57269 | 2.77 | 1.54 | – | – |
| QM F57270 | 2.91 | 1.59 | – | – |
| QM F57271 | 2.92 | 1.67 | – | – |
| QM F57272 | 2.71 | 1.59 | – | – |
| QM F57273 | – | – | 1.57 | 1.40 |
| QM F57274 | 2.83 | 1.62 | – | – |
| QM F57275 | 2.88 | 1.60 | – | – |
| QM F57276 | 3.22 | 1.50 | – | – |
| QM F57277 | 3.02 | 1.73 | – | – |
| QM F57278 | 3.01 | 1.74 | – | – |
| QM F57279 | 2.90 | 1.63 | – | – |
| QM F57280 | 2.87 | 1.67 | – | – |
| QM F57281 | 2.63 | 1.59 | – | – |
| QM F57282 | 2.76 | 1.52 | – | – |

**Notes.**
L, maximum length; W, maximum width.

and merged with T5 at most stages of wear. Posterior edge of T6 sweeps posteriorly slightly from T5 in most specimens. Posterior edge of T4-6 complex mostly arcuate anteriorly, enclosing T7-9 complex, especially on lingual side. T7 indistinguishable from T8 in the holotype but very small and completely incorporated into T8 in other specimens. T8 large and circular, orientated posteriorly. It is the highest cusp with all others roughly uniform in height. T9 small and elliptical. Lower half of T9 connects to T8 at early stages of wear, becoming fully incorporated with further wear.

With three roots. Anterior root the largest of the three. It is circular and directed anteriorly from the accessory cusp and T2. Posterolingual root narrow and plunges vertically from T1 and T4. Posterior root of equal size with posterolingual root, is more circular and is elongate vertically from T8 and T9.

**M$^2$**: Triangular in shape with broadest point along anterior edge. T1 circular and distinct, cusp directed posteriorly with occlusal surface inclined proximally. Deep furrows on buccal and posterior side of T1 separate it from other cusps and retains identity through wear. T2 and T3 absent. T4 large, elongated and tear-shaped, stretching posterolingually. Anterior edge of T4 sits posterior to posterior edge of T5. T4 posteriorly inclined, with occlusal surface facing proximally, similar to T1. T5 only slightly larger than T4 and subtriangular, connecting to T4 by its enamel rim and directed posteriorly. T6 absent. Posterior edge of T4-5 complex arcuate anteriorly, enclosing T8. T7 almost indistinguishable from T8 but indicated by a small bulge on the lingual edge of T8. T8 large and circular, directed posteriorly. Posterior edge arcuate posteriorly and delineates the most posterior edge of the tooth. No obvious indication of presence of T9. Remnant of furrow that marked its position present, indicating it has been wholly incorporated into T8. Enamel rim of cusps is variable, with T5 and T8 thicker than other cusps. All cusps of equal height and incline posteriorly at varying degrees, with T5 and T8 leaning posteriorly more than T1 and T4.

Roots not visible on only available specimen of M$^2$. Description has been gathered from alveoli in a specimen also preserving M$^1$ (QM F57275). M$^2$ has three roots. Lingual root very large and elongated, directed vertically. Anterobuccal root is circular, extends anteriorly, and is smaller than the lingual root. Posterobuccal root smallest of the three, elongated and extends vertically.

**M$^3$**: no specimen known.

Information on dental arcade is limited. Large posterior palatal foramen extends distally from posterior of M$^1$. Zygomatic plate wide with posterior edge appearing almost straight but is slightly convex.

# DISCUSSION

## Taphonomy

Even though Riversleigh's Rackham's Roost Site and Floraville's Site 5C represent vast differences in both mode of death and environment of preservation, similar skeletal elements have been preserved. Rackham's Roost Site is interpreted to have been a Ghost Bat (*Macroderma gigas*) roost during the Early Pleistocene (*Hand, 1996*; *Woodhead et al., in press*) and specimens of *Leggadina* found there are thought to be the result of bat

predation (*Godthelp, 1997*). Floraville's Site 5C specimens are more likely to have come from marsupial predators, fossils of which have also been found at the site (*Rich et al., 1991*). There have been no complete skulls found at either site. The fractured cranial and post cranial elements found cannot be attributed to individual murine taxa due to overlaps in size and a lack of features known to separate them (H Godthelp, 2013, unpublished data).

The Rackham's Roost assemblage contains only upper molars (upper = 28, lower = 0) which are all identifiable as belonging to the same species of *Leggadina*. They include complete molar rows, dental arcades and zygomatic plates. Site 5C specimens are dominated by lower molars but lack any molar rows (upper = 17, lower = 20). Upper molars all belong to the same species of *Leggadina*. Lower molars are observed to belong to the genus *Leggadina* but since they are not associated at all with the upper molars found, it is not possible to confidently identify them as belonging to the same species. Dental arcade and zygomatic plate information is fragmentary. An increased preservation of upper molars over lower molars is expected since the lower molars, attached to the mandible, have a greater chance of early disarticulation before preservation, whereas the upper molars are more likely to be retained in situ with the skull and post cranial bones for a longer period of time (*Behrensmeyer, 1984*). Nevertheless, it is important to note that the mandible tends to be stronger than the cranium, suggesting the large number of lower molars at Site 5C is the result of the lowers surviving the preservation process more readily than the uppers (*Behrensmeyer, 1984*). It is possible that sampling could have played a part in these results. The question then is whether further sampling at numerous places at Site 5C would increase the number of upper molars found. The only way to test this is through continued sampling. The Rackham's Roost specimens on the other hand would have suffered little disturbance during the process of fossilisation as specimens would have been protected inside the cave until it eroded. This is the likely reason more complete molar rows have been found at this site, however this does not explain why so few lower molars have been found. Again this could be due to sampling (*Lundelius, 2006*).

The occlusal surface of molars from specimens found at Rackham's Roost Site provides additional information on the age of individual animals through the degree of wear present on molars. The specimens collected from Rackham's Roost are dominated by largely unworn occlusal features, indicating a large number of the specimens were juveniles. *Macroderma gigas* moves to different feeding roosts to take advantage of seasonal resources, and it is likely they followed the breeding cycles of its prey, explaining the dominance of juveniles in the sample (*Tidemann et al., 1985*).

### Environmental impact

The early and middle Miocene in Australia was characterised by high levels of rainfall and the dominance of rainforest communities (*Martin, 2006*). As Australia moved from 'greenhouse' to 'icehouse' conditions in the later Miocene (10-5mya) the environment became increasingly arid and the biota needed to adapt (*Dawson & Dawson, 2006*). Environmental communities also changed during the Pliocene from rainforest dominated

areas to mosaics of grassland and open woodland (*Archer, Hand & Godthelp, 1991*). The changing distribution and diversity of mammals in the Riversleigh World Heritage Area fossil deposits is evidence of these changes (*Archer et al., 1989*; *Travouillon et al., 2009*). It is likely that as these changes occurred, arid-type responses were produced in much of its fauna (*Archer et al., 1998*), as seen in the Alcoota assemblage in the Northern Territory which shows a marked change in biota present in the late Miocene and early Pliocene (*Black et al., 2012*). By the end of the Pleistocene period animals of all types were forced to adapt their diet and behaviour where possible in order to survive because great climatic fluctuations caused by over 20 cycles of glacial and interglacial periods resulted in unpredictability of resources (*Archer et al., 1998*; *Martin, 2006*).

Continent-wide climatic shifts during the Pliocene and Pleistocene were very fast in terms of evolutionary response time, requiring taxa to either adapt quickly, be resilient enough to survive, or to be lost entirely (*Archer et al., 1998*). One of the factors that characterises the success of rodents in Australia is their rapid speciation (*Bush et al., 1977*). Modern *Leggadina* species inhabit arid-environments in northeastern Queensland (*L. lakedownensis*) and a variety of areas through inland Australia (*L. forresti*) (*Watts & Aslin, 1981*). However, the environment of Southeast Asia during the Miocene, thought to be the originating point of Australian murids, was characterised by tropical rainforests which were slowly beginning to contract (*Heaney , 1991*). It is therefore likely that the genus *Leggadina* evolved from an ancestor which was not arid-adapted.

Species of *Leggadina* have reasonably complex upper molars in comparison to closely related taxa, for example, both *Leggadina gregoriensis* and *L. macrodonta* have an additional occlusal structure (furrows) that allows for increased precision during mastication, indicating the evolution and specialisation of their teeth for a predominantly granivorous diet (*Herring, 1993*; *Evans et al., 2007*). Similarly, the width of the zygomatic plate is a useful indicator of the kinds of food eaten by rodents, because width of the zygomatic plate increases with an increase in the size of the anterior deep masseter muscle used for pulverising food (*Watts & Aslin, 1981*; *Satoh, 1997*). The zygomatic plate in both fossil species is quite wide suggesting further specialisation for a predominately granivorous diet. When the fossil *Leggadina* species evolved these adaptations cannot be determined at the moment due to the lack of knowledge on both the timing and method of their dispersal to and within Australia, as well as appropriate morphological evidence for other Australian fossil species.

One particularly interesting feature distinguishing *Leggadina macrodonta* is the size of its teeth, particularly $M^1$ which is up to 18% larger than the $M^1$ of *L. gregoriensis* or the two modern forms. The increase in size of the teeth and occlusal structures could be due to a number of different factors. Larger teeth would be a useful adaptation for taking advantage of a wider variety of resources necessary for survival in a changeable climate. Alternatively, increased tooth size could represent specialisation for a more selective diet, again resulting from a changing environment. It is also possible the increase in size of the molars of *L. macrodonta* was due to an increase in overall body mass, with this particular species growing larger in order to compete against larger animals for resources, as well as

becoming able to process low nutrient foods more easily and reduce water loss (*Archer et al., 1998*; *Dawson & Dawson, 2006*). Unfortunately it is not possible to calculate body mass of this species currently due to the absence of adequate lower molar data and a lack of long bones in the fossil assemblage relatable to this species (*Hopkins , 2008*). However, the question that remains is why did *L. macrodonta* develop exceptionally large molars, while molar size in *L. gregoriensis* remained more closely aligned with its modern relatives. Broader ecological evidence needs to be presented on changes in tooth structure in other species during the Plio-Pleistocene to make a more informed determination on tooth variation between *L. macrodonta* and *L. gregoriensis*.

### Future work

Molar morphology has been an important tool for understanding the evolution of the Murinae and other rodent groups for over 100 years. At this point in time it is still essential for the description of new fossil species of Australian murids. However, to date there has been no comprehensive phylogenetic analysis based on morphology including both fossil and modern species. The leading analysis on morphological relationships using molar morphology was conducted over 40 years ago (*Misonne, 1969*). On the other hand, advances in molecular assessment of murid relationships have proliferated over the past 30 years (*Baverstock et al., 1981*; *Pascale, Valle & Furano, 1990*; *Catzeflis, Aguilar & Jaeger, 1992*; *Watts et al., 1992*; *Jansa & Weksler, 2004*; *Steppan et al., 2005*; *Rowe et al., 2008*; *Nilsson et al., 2010*; *Schenk, Rowe & Steppan, 2013*). An updated morphological phylogeny combined with molecular phylogenies would give a much more cohesive picture of Australian murid evolutionary history than using either alone (*Wiens, 2004*; *Aplin, 2006*).

## CONCLUSION

Murid rodents are speciose in Australia, but their evolutionary relationships and origins have been shrouded in mystery due in large part to the paucity of fossil evidence available. Two new species of the genus *Leggadina*: *Leggadina gregoriensis* from the Pleistocene Rackham's Roost Site in the Riversleigh World Heritage Area and *Leggadina macrodonta* from the Plio-Pleistocene Site 5C at Floraville Station, both in northwestern Queensland, have been described here. Their description extends the temporal range of the genus *Leggadina* to around 2.5 million years. Both fossil species display increased complexity in the upper molars and larger attachment sites on the zygomatic plate, likely due to the development of a predominately granivorous diet. *L. macrodonta* also displays an increase in size of $M^1$ and $M^2$ which may be the result of a number of factors including adaptation to the unpredictability of, and increased competition for, resources in a changing climate or an increase in body size. Further research is essential to further develop understanding on the relationships and evolution of the genus *Leggadina* as well as the broader Murinae group.

## ACKNOWLEDGEMENTS

Thanks firstly to Mike Archer and Sue Hand for their supervision of this project and for their continued advice throughout the process. For access to specimens  I thank Sandy

Ingleby and Anja Divljan from the Australian Museum. Thanks also to Anna Gillespie for preparation of specimens and assistance with sorting and numbering fossil specimens, and Troy Myers for assistance with PAST.

### Funding
The authors declare there was no funding for this work.

### Competing Interests
The authors declare there are no competing interests.

### Author Contributions
- Ada J. Klinkhamer performed the experiments, analyzed the data, wrote the paper, prepared figures and/or tables, reviewed drafts of the paper.
- Henk Godthelp conceived and designed the experiments, contributed reagents/materials/analysis tools.

### New Species Registration
The following information was supplied regarding the registration of a newly described species:

macrodonta: http://zoobank.org/NomenclaturalActs/9B165602-2333-4669-8716-23151970F9CF

gregoriensis: http://zoobank.org/NomenclaturalActs/22FAC03C-433A-4D00-9E51-FEBDF3DCBB67.

### Supplemental Information
Supplemental information for this article can be found online at http://dx.doi.org/10.7717/peerj.1088#supplemental-information.

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
