# Peer review of "Two new species of fossil Leggadina (Rodentia: Muridae) from Northwestern Queensland"

_PeerJ, doi:10.7717/peerj.1088_

## Round 0.1 · original submission · Major Revisions

I've read and considered all of the reviewers' suggestions and I find them to be reasonable (with one exception, as discussed below). In particular, I agree with Reviewer 1 and Reviewer 3 that the lower jaws and teeth should be included in the descriptions of the new taxa. This is something that will need to be done eventually anyway, and including those descriptions in this manuscript will only strengthen it and make it more useful for future workers.

Please give special attention to the following points:
- The presence or absence of the M1 accessory cusp in the L. gregoriensis paratype, and the suitability of that specimen for paratype status if it is lacking an important diagnostic feature. (Reviewer 1)
- The dating of Rackham's Roost and its implications for the temporal range of Leggadina. (Reviewer 2)
- The quality of Figure 4B and 4C could be improved. (Reviewer 2)

This is not an exhaustive list, just three of the most important improvmements suggested by the reviewers. In particular, the annotated PDF provided by Reviewer 3 has many other suggestions for improvement.

One change you do not need to make is to find a new specific epithet for Leggadina macrodonta, as suggested by Reviewer 3. The species name 'macrodonta' is in use for a spider in the genus Solenysa (Wang, Fang, Hirotsugu Ono & Lihong Tu. 2015 A review of Solenysa spiders from Japan (Araneae, Linyphiidae), with a comment on the type species S. mellotteei Simon, 1894. ZooKeys 481: 39-56) - but that is no barrier to using it here. The ICZN only requires that genus names be unique, whereas species names can be, and are, reused, especially if the taxa are so distantly related (e.g., rodents and spiders) that no future synonymization problems will arise. For example, the species name 'cuvieri' is used for a gazelle (Gazella), a rail (Dryolimnas), and a fish (Orestias).

All three of the reviewers found value in your work and I hope that you take their suggestions for improvement in the constructive spirit in which they are offered.

·

Basic reporting

The writing is fairly clear, sufficient background information is provided, figures are good quality and adequate (but see below). The very first reference cited (Lines 11 and 14: Godthelp 2001) is not listed in the References.
Lines 60 and 63: Godthelp is one of the authors, so citing his personal communications would not seem to be necessary, but given that the information/interpretation provided in these sentences is unpublished, should these maybe be cited as "Godthelp unpublished data" instead? [I'm not sure what the convention is for this kind of thing in PeerJ]

Experimental design

The fossil specimens are preserved in a publicly accessible collection, the Queensland Museum, which is an important technical standard and affords further study and potential reproducibility by other investigators. Descriptive and statistical treatments are standard practice and are suitable.

In the Differential Diagnosis paragraph the authors state (Line 140) that all their Leggadina specimens show the anterior M1 accessory cusp considered typical for the genus. However the paratype (QM F57244) of L. gregoriensis shown in Fig. 6 lacks this cusp although the other specimens shown have it. Is it simply worn away in this paratype specimen? Maybe one of the other more complete specimens listed would better serve as a paratype if they show this cusp?

One concern with the Systematic Paleontology section is that only maxillae and upper teeth are studied and described. Later in the manuscript in the taphonomic Discussion (lines 360-363) the authors indicate that numerous lower molars and dentaries were found and were actually more common than uppers at Floraville Site 5C (but no lowers were recovered at Rackham's Roost). Dentaries and lower teeth can provide additional systematic characters for distinguishing various murine species. Yet, they are not included in the species description of Leggadina macrodonta from Floraville. Although they are disarticulated and thus direct association of lowers with uppers is lost, lower jaw elements can sometimes be associated with uppers through other means such as size, proportions or qualitative features that might be expected to be correlated between upper and lower molars. Their inclusion in the description of L. macrodonta might aid in the association of lower teeth of the other species L. gregoriensis, too, if lowers of that species are eventually discovered at Rackham's Roost. This omission of lower jaws and teeth is not necessarily a critical deficit in the manuscript, but their inclusion if possible is standard practice and could certainly enhance the one species' Diagnosis. If not included in the diagnosis/description, at least a sentence or two explaining the authors' rationale for excluding the lowers from the species descriptions, especially for L. macrodonta, maybe in the introduction to the Systematic Paleo section, would be welcome. As noted by the authors themselves in the subsection "Future Work" (Lines 451-461), the kind of morphological descriptions of molars as included in this manuscript are necessary and important for understanding the evolution of Australian murine rodents. I agree with this statement and would argue that including the lowers in the diagnosis of L. macrodonta could aid in this process, too. Unfortunately, it would also necessitate including the terminology for lower molars in Figure 1 and illustrating the lowers together with the uppers for the relevant new species in other existing figures or adding new figures.

In the Description sections for each new species, there are duplicated paragraphs for each of the teeth. The first series of these for each new species describes the crowns for each molar with paragraphs subheaded as M1, M2, and M3 (Lines 208, 224, 236); the second series describes the roots for each, again listed as M1, M2, and M3 (Lines .248, 252, 255). These duplications seem unnecessary and in fact on first view it is a little confusing to see these paragraphs listed twice with the same subheadings (M1, M2, M3) . Why not simply combine the roots description of each tooth with the crown description for each?

Validity of the findings

The data and conclusions are good as is, and provide a solid basis for establishing the two new species of Leggadina. As such it is not absolutely critical to include the lower jaws and molars in the hypodigm and diagnosis for one of the new species, L. macrodonta. However, doing so would strengthen the description of L. macrodonta, as noted above, and also make widely available a more complete picture of this taxon. In other words, if you have the specimens in hand, why not include them?

Additional comments

No other comments.

·

Basic reporting

The article is clearly written and largely free of typos, as far as I could find.
Most of the relevant literature, with one important exception, is cited (see comments in 'validity of the findings' section).
The figures are necessary and sufficient and the SEM images appear to be of high quality.
I would raise the issue of the quality of the reproduction of the drawings in Figure 4, parts B and C. I know that review pdfs tend to produce washed out figures that look much better in the final publication but I think these pencil drawings are not of sufficient quality, they are a bit sketchy and rough in places and the pencil outlines are too faint. I would recommend that these figures are redrafted in ink or on with a computer graphics package, in the style of figure 1. This should not be an onerous task.

Experimental design

No comments.

Validity of the findings

Although I am not an expert in rodent taxonomy, the species do seem to be well supported and there placement in Leggadina supported, at least on the grounds of raw similarity rather than synapomorphies. I do note that the dignoses lack any sort of discussion of what character states are thought to be derived within Leggadina and which are thought to be plesiomorphies. It may be that Australian murine phylogeny is so uncertain it is not possible to map character states at all, if this is the case it needs to be stated.
One major omission is any mention of the recent radiometric dating of Rackham's Roost site in:

Woodhead, J., Hand, S., Archer, M., Graham, I., Sniderman, K., Arena, D. A., Black, K., Godthelp, H., Creaser, P., Price, E. 2014. Developing a radiometrically-dated chronologic sequence for Neogene biotic change in Australia, from the Riversleigh World Heritage Area of Queensland. Gondwana Research (2014), doi: 10.1016/j.gr.2014.10.004

Woodhead et al. is available as a pre-print and citable, if not officially published yet. Its ommission from the present MS is is somewhat puzzling given that the junior co-author of this paper is one of the authors on Woodhead et al. Maybe there has been a long delay between writing this paper and its submission.
Anyway Woodhead find an Early Pleistocene age (2.1 ma) for a flowstone immediately overlying the fossiliferous layer. Although this is a minimal age, Woodhead argue that it more accurately reflects the age of the fossil deposit than biocorrelation does. Certainly an early Pliocene age of 5 million years for Rackham's Roost cited in this paper would seem most unlikley (particularly since other deposits of similar antiquity such as Hamilton and Corra Lynn Cave are devoid of rodents). If the authors do not accept the radiometric date, that is fine but it should still be cited and discusssed with their reasons for not accepting the date. If an Early Pleistocene date for Rackham's Roost is accepted then the paper's conclusions about a long temporal range for the genus Leggadina fall away and the text needs to be modified accordingly.

Additional comments

I have appended some comments to a pdf of the MS.

·

Basic reporting

See attached.

Experimental design

See attached.

Validity of the findings

See attached.

Additional comments

See attached.

---

## Round 0.2 · accepted · Accept

I have reviewed the revised manuscript and your rebuttal letter. The one major thing that the reviewers had asked for was a description of the lower jaw material. Reviewer 3 felt most strongly about this. Unfortunately he is currently out in the field and has not had a chance to review the revised manuscript. However, I was able to converse with him briefly over email and I conveyed your explanation, that the lower jaw material could not be confidently associated with the cranial material. He found that explanation satisfactory. In my view, all issues with the manuscript have been satisfactorily resolved, and I am happy to accept it for publication in PeerJ.

The decision of whether or not to publish the peer reviews alongside the paper is entirely yours, and will not affect how your paper is handled going forward. However, I encourage you to do so. All of the reviewers chose to sign their reviews, and making the reviews public allows them to receive more credit for their efforts, and also contributes to the emerging culture of fairness and transparency in editing and peer review.